# GRAPH NEURAL NETWORKS WITH GENERATED PARAMETERS FOR RELATION EXTRACTION

## ABSTRACT

Recently, progress has been made towards improving relational reasoning in machine learning field. Among existing models, graph neural networks (GNNs) is one of the most effective approaches for multi-hop relational reasoning. In fact, multi-hop relational reasoning is indispensable in many natural language processing tasks such as relation extraction. In this paper, we propose to generate the parameters of graph neural networks (GP-GNNs) according to natural language sentences, which enables GNNs to process relational reasoning on unstructured text inputs. We verify GP-GNNs in relation extraction from text. Experimental results on a human-annotated dataset and two distantly supervised datasets show that our model achieves significant improvements compared to baselines. We also perform a qualitative analysis to demonstrate that our model could discover more accurate relations by multi-hop relational reasoning.

## 1 INTRODUCTION

Recent years, graph neural networks (GNNs) have been applied to various fields of machine learning, including node classification (Kipf & Welling, 2016), relation classification (Schlichtkrull et al., 2017), molecular property prediction (Gilmer et al., 2017), few-shot learning (Garcia & Bruna, 2018), and achieve promising results on these tasks. These works have demonstrated GNNs' strong power to process relational reasoning on graphs.

Relational reasoning aims to abstractly reason about entities/objects and their relations, which is an important part of human intelligence. Besides graphs, relational reasoning is also of great importance in many natural language processing tasks such as question answering, relation extraction, summarization, etc. Consider the example shown in Fig. 1, existing relation extraction models could easily extract the facts that *Luc Besson* directed a film *Léon: The Professional* and that the film is in *English*, but fail to infer the relationship between *Luc Besson* and *English* without multi-hop relational reasoning. By considering the reasoning patterns, one can discover that *Luc Besson* could speak *English* following a reasoning logic that *Luc Besson* directed *Léon: The Professional* and this film is in *English* indicates *Luc Besson* could speak *English*. However, most existing GNNs can only process multi-hop relational reasoning on pre-defined graphs and cannot be directly applied in natural language relational reasoning. Enabling multi-hop relational reasoning in natural languages remains an open problem.

To address this issue, in this paper, we propose graph neural networks with generated parameters (GP-GNNs), to adapt graph neural networks to solve the natural language relational reasoning task. GP-GNNs first constructs a fully-connected graph with the entities in the sequence of text. After that, it employs three modules to process relational reasoning: (1) an encoding module which enables edges to encode rich information from natural languages, (2) a propagation module which propagates relational information among various nodes, and (3) a classification module which makes predictions with node representations. As compared to traditional GNNs, GP-GNNs could learn edges' parameters from natural languages, extending it from performing inferring on only non-relational graphs or graphs with a limited number of edge types to unstructured inputs such as texts.

In the experiments, we apply GP-GNNs to a classic natural language relational reasoning task: relation extraction from text. We carry out experiments on Wikipedia corpus aligned with Wikidata knowledge base (Vrandečić & Krötzsch, 2014) and build a human annotated test set as well as two distantly labeled test sets with different levels of denseness.Experiment results show that our

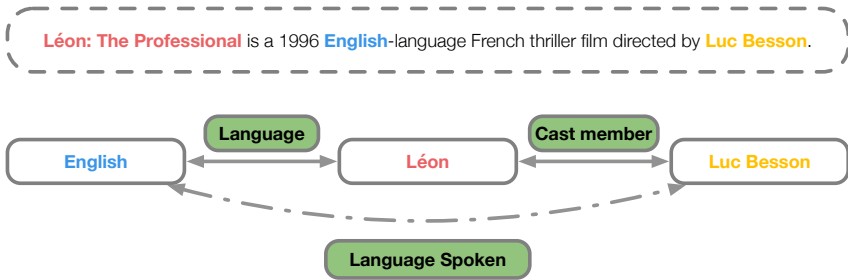

Figure 1: An example of relation extraction from plain text. Given a sentence with several entities marked, we model the interaction between these entities by generating the weights of graph neural networks. Modeling the relationship between "Léon" and "English" as well as "Luc Besson" helps discover the relationship between "Luc Besson" and "English".

model outperforms other state-of-the-art models on relation extraction task by considering multi-hop relational reasoning. We also perform a qualitative analysis which shows that our model could discover more relations by reasoning more robustly as compared to baseline models.

Our main contributions are in two-fold:

(1) We extend a novel graph neural network model with generated parameters, to enable relational message-passing with rich text information, which could be applied to process relational reasoning on unstructured inputs such as natural languages.

(2) We verify our GP-GNNs in the task of relation extraction from text, which demonstrates its ability on multi-hop relational reasoning as compared to those models which extract relationships separately. Moreover, we also present three datasets, which could help future researchers compare their models in different settings.

## 2 RELATED WORK

### 2.1 GRAPH NEURAL NETWORKS (GNNs)

GNNs were first proposed in (Scarselli et al., 2009) and are trained via the Almeida-Pineda algorithm (Almeida, 1987). Later the authors in Li et al. (2016) replace the Almeida-Pineda algorithm with the more generic backpropagation and demonstrate its effectiveness empirically. Gilmer et al. (2017) propose to apply GNNs to molecular property prediction tasks. Garcia & Bruna (2018) shows how to use GNNs to learn classifiers on image datasets in a few-shot manner. Gilmer et al. (2017) study the effectiveness of message-passing in quantum chemistry. Dhingra et al. (2017) apply message-passing on a graph constructed by coreference links to answer relational questions. There are relatively fewer papers discussing how to adapt GNNs to natural language tasks. For example, Marcheggiani & Titov (2017) propose to apply GNNs to semantic role labeling and Schlichtkrull et al. (2017) apply GNNs to knowledge base completion tasks. Zhang et al. (2018) apply GNNs to relation extraction by encoding dependency trees, and De Cao et al. (2018) apply GNNs to multi-hop question answering by encoding co-occurence and co-reference relationships. Although they also consider applying GNNs to natural language processing tasks, they still perform message-passing on predefined graphs. Johnson (2017) introduces a novel neural architecture to generate a graph based on the textual input and dynamically update the relationship during the learning process. In sharp contrast, this paper focuses on extracting relations from real-world relation datasets.

### 2.2 RELATIONAL REASONING

Relational reasoning has been explored in various fields. For example, Santoro et al. (2017) propose a simple neural network to reason the relationship of objects in a picture, Xu et al. (2017) build up a scene graph according to an image, and Kipf et al. (2018) model the interaction of physical objects.

In this paper, we focus on the relational reasoning in natural language domain. Existing works (Zeng et al., 2014; 2015; Lin et al., 2016) have demonstrated that neural networks are capable of capturing

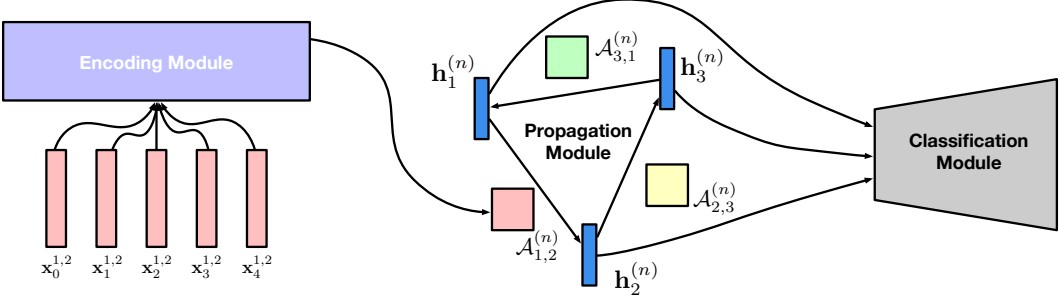

Figure 2: Overall architecture: the encoding module takes a sequence of vector representations as inputs, and output a transition matrix as output; the propagation module propagates the hidden states from nodes to its neighbours with the generated transition matrix; the classification module provides task-related predictions according to nodes representations.

the pair-wise relationship between entities in certain situations. For example, (Zeng et al., 2014) is one of the earliest works that applies a simple CNN to this task, and (Zeng et al., 2015) further extends it with piece-wise max-pooling. Nguyen & Grishman (2015) propose a multi-window version of CNN for relation extraction. Lin et al. (2016) study an attention mechanism for relation extraction tasks. Peng et al. (2017) predict n-ary relations of entities in different sentences with Graph LSTMs. Le & Titov (2018) treat relations as latent variables which are capable of inducing the relations without any supervision signals. Zeng et al. (2017) show that the relation path has an important role in relation extraction. Miwa & Bansal (2016) show the effectiveness of LSTMs (Hochreiter & Schmidhuber, 1997) in relation extraction. Christopoulou et al. (2018) proposed a walk-based model to do relation extraction. The most related work is (Sorokin & Gurevych, 2017), where the proposed model incorporates contextual relations with attention mechanism when predicting the relation of a target entity pair. The drawback of existing approaches is that they could not make full use of the multi-hop inference patterns among multiple entity pairs and their relations within the sentence.

## 3 GRAPH NEURAL NETWORK WITH GENERATED PARAMETERS (GP-GNNS)

We first define the task of natural language relational reasoning. Given a sequence of text with $m$ entities, it aims to reason on both the text and entities and make a prediction of the labels of the entities or entity pairs.

In this section, we will introduce the general framework of GP-GNNs. GP-GNNs first build a fully-connected graph $\mathcal{G} = (\mathcal{V}, \mathcal{E})$, where $\mathcal{V}$ is the set of entities, and each edge $(v_i, v_j) \in \mathcal{E}, v_i, v_j \in \mathcal{V}$ corresponds to a sequence $s = x_0^{i,j}, x_1^{i,j}, \ldots, x_{l-1}^{i,j}$ extracted from the text. After that, GP-GNNs employ three modules including (1) encoding module, (2) propagation module and (3) classification module to proceed relational reasoning, as shown in Fig. 2.

### 3.1 ENCODING MODULE

The encoding module converts sequences into transition matrices corresponding to edges, i.e. the parameters of the propagation module, by

$$\mathcal{A}_{i,j}^{(n)} = f(E(x_0^{i,j}), E(x_1^{i,j}), \cdots, E(x_{l-1}^{i,j}); \theta_e^n), \tag{1}$$

where $f(\cdot)$ could be any model that could encode sequential data, such as LSTMs, GRUs, CNNs, $E(\cdot)$ indicates an embedding function, and $\theta_e^n$ denotes the parameters of the encoding module of $n$-th layer.

### 3.2 PROPAGATION MODULE

The propagation module learns representations for nodes layer by layer. The initial embeddings of nodes, i.e. the representations of layer 0, are task-related, which could be embeddings that encode

features of nodes or just one-hot embeddings. Given representations of layer $n$, the representations of layer $n + 1$ are calculated by

$$\mathbf{h}_i^{(n+1)} = \sum_{v_j \in \mathcal{N}(v_i)} \sigma(\mathcal{A}_{i,j}^{(n)} \mathbf{h}_j^{(n)}), \tag{2}$$

where $\mathcal{N}(v_i)$ denotes the neighbours of node $v_i$ in graph $\mathcal{G}$ and $\sigma(\cdot)$ denotes non-linear activation function.

### 3.3 CLASSIFICATION MODULE

Generally, the classification module takes node representations as inputs and outputs predictions. Therefore, the loss of GP-GNNs could be calculated as

$$\mathcal{L} = g(\mathbf{h}_{0:|\mathcal{V}|-1}^0, \mathbf{h}_{0:|\mathcal{V}|-1}^1, \dots, \mathbf{h}_{0:|\mathcal{V}|-1}^K, Y; \theta_c), \tag{3}$$

where $\theta_c$ denotes the parameters of the classification module, $K$ is the number of layers in propagation module and $Y$ denotes the ground truth label. The parameters in GP-GNNs are trained by gradient descent methods.

## 4 RELATION EXTRACTION WITH GP-GNNS

Relation extraction from text is a classic natural language relational reasoning task. Given a sentence $s = (x_0, x_1, \dots, x_{l-1})$, a set of relations $\mathcal{R}$ and a set of entities in this sentence $\mathcal{V}_s = \{v_1, v_2, \dots, v_{|\mathcal{V}_s|}\}$, where each $v_i$ consists of one or a sequence of tokens, relation extraction from text is to identify the pairwise relationship $r_{v_i,v_j} \in \mathcal{R}$ between each entity pair $(v_i, v_j)$.

In this section, we will introduce how to apply GP-GNNs to relation extraction.

### 4.1 ENCODING MODULE

To encode the context of entity pairs (or edges in the graph), we first concatenate the position embeddings with word embeddings in the sentence:

$$E(x_t^{i,j}) = [\boldsymbol{x}_t; \boldsymbol{p}_t^{i,j}], \tag{4}$$

where $\boldsymbol{x}_t$ denotes the word embedding of word $x_t$ and $\boldsymbol{p}_t^{i,j}$ denotes the position embedding of word position $t$ relative to the entity pair's position $i, j$ (Details of these two embeddings are introduced in the next two paragraphs.) After that, we feed the representations of entity pairs into encoder $f(\cdot)$ which contains a bi-directional LSTM and a multi-layer perceptron:

$$\mathcal{A}_{i,j}^{(n)} = [\texttt{MLP}_n(\texttt{BiLSTM}_n((E(x_0^{i,j}), E(x_1^{i,j}), \cdots, E(x_{l-1}^{i,j}))], \tag{5}$$

where $n$ denotes the index of layer [1], $[\cdot]$ means reshaping a vector as a matrix, $\texttt{BiLSTM}$ encodes a sequence by concatenating tail hidden states of the forward LSTM and head hidden states of the backward LSTM together and $\texttt{MLP}$ denotes a multi-layer perceptron with non-linear activation $\sigma$.

**Word Representations** We first map each token $x_t$ of sentence $\{x_0, x_1, \dots, x_{l-1}\}$ to a $k$-dimensional embedding vector $\boldsymbol{x}_t$ using a word embedding matrix $W_e \in \mathbb{R}^{|V| \times d_w}$, where $|V|$ is the size of the vocabulary. Throughout this paper, we stick to 50-dimensional GloVe embeddings pre-trained on a 6 billion corpus (Pennington et al., 2014).

**Position Embedding** In this work, we consider a simple entity marking scheme[2]: we mark each token in the sentence as either belonging to the first entity $v_i$, the second entity $v_j$ or to neither of those. Each position marker is also mapped to a $d_p$-dimensional vector by a position embedding matrix $\boldsymbol{P} \in \mathbb{R}^{3 \times d_p}$. We use notation $\boldsymbol{p}_t^{i,j}$ to represent the position embedding for $x_t$ corresponding to entity pair $(v_i, v_j)$.

---

[1] Adding index to neural models means their parameters are different among layers.

[2] As pointed out by Sorokin & Gurevych (2017), other position markers lead to no improvement in performance.

## 4.2 PROPAGATION MODULE

Next, we use Eq. (2) to propagate information among nodes where the initial embeddings of nodes and number of layers are further specified as follows.

**The Initial Embeddings of Nodes** Suppose we are focusing on extracting the relationship between entity $v_i$ and entity $v_j$, the initial embeddings of them are annotated as $\mathbf{h}_{v_i}^{(0)} = \boldsymbol{a}_{\text{subject}}$, and $\boldsymbol{h}_{v_j}^{(0)} = \boldsymbol{a}_{\text{object}}$, while the initial embeddings of other entities are set to all zeros. We set special values for the head and tail entity's initial embeddings as a kind of "flag" messages which we expect to be passed through propagation. Annotators $\boldsymbol{a}_{\text{subject}}$ and $\boldsymbol{a}_{\text{object}}$ could also carry the prior knowledge about subject entity and object entity. In our experiments, we generalize the idea of Gated Graph Neural Networks (Li et al., 2016) by setting $\boldsymbol{a}_{\text{subject}} = [\mathbf{1}; \mathbf{0}]^\top$ and $\boldsymbol{a}_{\text{object}} = [\mathbf{0}; \mathbf{1}]^{\top\,3}$.

**Number of Layers** In general graphs, the number of layers $K$ is chosen to be of the order of the graph diameter so that all nodes obtain information from the entire graph. In our context, however, since the graph is densely connected, the depth is interpreted simply as giving the model more expressive power. We treat $K$ as a hyper-parameter, the effectiveness of which will be discussed in detail (Sect. 5.4).

## 4.3 CLASSIFICATION MODULE

The output module takes the embeddings of the target entity pair $(v_i, v_j)$ as input, which are first converted by:

$$\boldsymbol{r}_{v_i,v_j} = [[\boldsymbol{h}_{v_i}^{(1)} \odot \boldsymbol{h}_{v_j}^{(1)}]^\top ; [\boldsymbol{h}_{v_i}^{(2)} \odot \boldsymbol{h}_{v_j}^{(2)}]^\top ; \ldots ; [\boldsymbol{h}_{v_i}^{(K)} \odot \boldsymbol{h}_{v_j}^{(K)}]^\top], \tag{6}$$

where $\odot$ represents element-wise multiplication. This could be used for classification:

$$\mathbb{P}(r_{v_i,v_j}|h,t,s) = \texttt{softmax}(\texttt{MLP}(\boldsymbol{r}_{v_i,v_j})), \tag{7}$$

where $r_{v_i,v_j} \in \mathcal{R}$, and $\texttt{MLP}$ denotes a multi-layer perceptron module.

We use cross entropy here as the classification loss

$$\mathcal{L} = \sum_{s \in S} \sum_{i \neq j} \log \mathbb{P}(r_{v_i,v_j}|i,j,s), \tag{8}$$

where $r_{v_i,v_j}$ denotes the relation label for entity pair $(v_i, v_j)$ and $S$ denotes the whole corpus.

In practice, we stack the embeddings for every target entity pairs together to infer the underlying relationship between each pair of entities. We use PyTorch (Paszke et al., 2017) to implement our models. To make it more efficient, we avoid using loop-based, scalar-oriented code by matrix and vector operations.

# 5 EXPERIMENTS

Our experiments mainly aim to: (1) showing that our best models could improve the performance of relation extraction under a variety of settings; (2) illustrating that how the number of layers affect the performance of our model; and (3) performing a qualitative investigation to highlight the difference between our models and baseline models. In both part (1) and part (2), we do three subparts of experiments: (i) we will first show that our models could improve instance-level relation extraction on a human annotated test set, and (ii) then we will show that our models could also help enhance the performance of bag-level relation extraction on a distantly labeled test set [4], and (iii) we also split a subset of distantly labeled test set, where the number of entities and edges is large.

---

[3]The dimensions of $\mathbf{1}$ and $\mathbf{0}$ are the same. Hence, $d_r$ should be positive even integers. The embedding of subject and object could also carry the type information by changing annotators. We leave this extension for future work.

[4]Bag-level relation extraction is a widely accepted scheme for relation extraction with distant supervision, which means the relation of an entity pair is predicted by aggregating a bag of instances.

### 5.1 EXPERIMENT SETTINGS

#### 5.1.1 DATASETS

**Distantly labeled set** Sorokin & Gurevych (2017) have proposed a dataset with Wikipedia corpora. There is a small difference between our task and theirs: our task is to extract the relationship between every pair of entities in the sentence, whereas their task is to extract the relationship between the given entity pair and the context entity pairs. Therefore, we need to modify their dataset: (1) We added reversed edges if they are missing from a given triple, e.g. if triple (Earth, `part of`, Solar System) exists in the sentence, we add a reversed label, (Solar System, `has a member`, Earth), to it; (2) For all of the entity pairs with no relations, we added "NA" labels to them.[5] We use the same training set for all of the experiments.

**Human annotated test set** Based on the test set provided by (Sorokin & Gurevych, 2017), 5 annotators[6] are asked to label the dataset. They are asked to decide whether or not the distant supervision is right for every pair of entities. Only the instances accepted by all 5 annotators are incorporated into the human annotated test set. There are 350 sentences and 1,230 triples in this test set.

**Dense distantly labeled test set** We further split a dense test set from the distantly labeled test set. Our criteria are: (1) the number of entities should be strictly larger than 2; and (2) there must be at least one circle (with at least three entities) in the ground-truth label of the sentence [7]. This test set could be used to test our methods' performance on sentences with the complex interaction between entities. There are 1,350 sentences and more than 17,915 triples and 7,906 relational facts in this test set.

#### 5.1.2 MODELS FOR COMPARISON

We select the following models for comparison, the first four of which are our baseline models.

**Context-Aware RE**, proposed by Sorokin & Gurevych (2017). This model utilizes attention mechanism to encode the context relations for predicting target relations. It was the state-of-the-art models on Wikipedia dataset. This baseline is implemented by ourselves based on authors' public repo[8].

**Multi-Window CNN**. Zeng et al. (2014) utilize convolutional neural networks to classify relations. Different from the original version of CNN proposed in (Zeng et al., 2014), our implementation, follows (Nguyen & Grishman, 2015), concatenates features extracted by three different window sizes: 3, 5, 7.

**PCNN**, proposed by Zeng et al. (2015). This model divides the whole sentence into three pieces and applies max-pooling after convolution layer piece-wisely. For CNN and following PCNN, the entity markers are the same as originally proposed in (Zeng et al., 2014; 2015).

**LSTM or GP-GNN with $K = 1$ layer**. Bi-directional LSTM (Schuster & Paliwal, 1997) could be seen as an 1-layer variant of our model.

**GP-GNN with $K = 2$ or $K = 3$ layerss**. These models are capable of performing 2-hop reasoning and 3-hop reasoning, respectively.

#### 5.1.3 HYPER-PARAMETERS

We select the best parameters for the validation set. We select non-linear activation functions between `relu` and `tanh`, and select $d_n$ among $\{2, 4, 8, 12, 16\}$[9]. We have also tried two forms of adjacent matrices: tied-weights (set $\mathcal{A}^{(n)} = \mathcal{A}^{(n+1)}$) and untied-weights. Table 1 shows our best hyper-parameter settings, which are used in all of our experiments.

---

[5]We also resolve entities at the same position and remove self-loops from the previous dataset. Furthermore, we limit the number of entities in one sentence to 9, resulting in only 0.0007 data loss.

[6]They are all well-educated university students.

[7]Every edge in the circle has a non-"NA" label.

[8]`https://github.com/UKPLab/emnlp2017-relation-extraction`

[9]We set all $d_n$s to be the same as we do not see improvements using different $d_n$s

| Hyper-parameters | Value |
|---|---|
| learning rate | 0.001 |
| batch size | 50 |
| dropout ratio | 0.5 |
| hidden state size | 256 |
| non-linear activation $\sigma$ | relu |
| embedding size for #layers = 1 | 8 |
| embedding size for #layers = 2 and 3 | 12 |
| adjacent matrices | untied |

Table 1: Hyper-parameters settings.

## 5.2 EVALUATION DETAILS

So far, we have only talked about the way to implement sentence-level relation extraction. To evaluate our models and baseline models in bag-level, we utilize a bag of sentences with given entity pair to score the relations between them. Zeng et al. (2015) formalize the bag-level relation extraction as multi-instance learning. Here, we follow their idea and define the score function of entity pair and its corresponding relation $r$ as a max-one setting:

$$E(r|v_i, v_j, S) = \max_{s \in S} \mathbb{P}(r_{v_i, v_j}|i, j, s).$$

(9)

| Dataset | Human Annotated Test Set | |
|---|---|---|
| Metric | Acc | Macro F1 |
| Multi-Window CNN | 47.3 | 17.5 |
| PCNN | 30.8 | 3.2 |
| Context-Aware RE | 68.9 | 44.9 |
| GP-GNN (#layers=1) | 62.9 | 44.1 |
| GP-GNN (#layers=2) | 69.5 | 44.2 |
| GP-GNN (#layers=3) | **75.3** | **47.9** |

Table 2: Results on human annotated dataset

| Dataset | Distantly Labeled Test Set | | | | Dense Distantly Labeled Test Set | | | |
|---|---|---|---|---|---|---|---|---|
| Metric | P@5% | P@10% | P@15% | P@20% | P@5% | P@10% | P@15% | P@20% |
| Multi-Window CNN | 78.9 | 78.4 | 76.2 | 72.9 | 86.2 | 83.4 | 81.4 | 79.1 |
| PCNN | 73.0 | 65.4 | 58.1 | 51.2 | 85.3 | 79.1 | 72.4 | 68.1 |
| Context-Aware RE | 90.8 | 89.9 | 88.5 | 87.2 | 93.5 | 93.0 | 93.8 | 93.0 |
| GP-GNN (#layers=1) | 90.5 | 89.9 | 88.2 | 87.2 | 97.4 | 93.5 | 92.4 | 91.9 |
| GP-GNN (#layers=2) | 92.5 | **92.0** | 89.3 | 87.1 | 95.0 | 94.6 | 95.2 | 94.2 |
| GP-GNN (#layers=3) | **94.2** | **92.0** | **89.7** | **88.3** | **98.5** | **97.4** | **96.6** | **96.1** |

Table 3: Results on distantly labeled test set

## 5.3 EFFECTIVENESS OF REASONING MECHANISM

From Table 2 and 3, we can see that our best models outperform all the baseline models significantly on all three test sets. These results indicate our model could successfully conduct reasoning on the fully-connected graph with generated parameters from natural language. These results also indicate that our model not only performs well on sentence-level relation extraction but also improves on bag-level relation extraction. Note that Context-Aware RE also incorporates context information to predict the relation of the target entity pair, however, we argue that Context-Aware RE only models the co-occurrence of various relations, ignoring whether the context relation participates in the reasoning process of relation extraction of the target entity pair. Context-Aware RE may introduce more noise, for it may mistakenly increase the probability of a relation with the similar topic with the context relations. We will give samples to illustrate this issue in Sect. 5.5. Another interesting observation is that our #layers=1 version outperforms CNN and PCNN in these three datasets. One probable reason is that sentences from Wikipedia corpus are always complex, which

may be hard to model for CNN and PCNN. Similar conclusions are also reached by Zhang & Wang (2015).

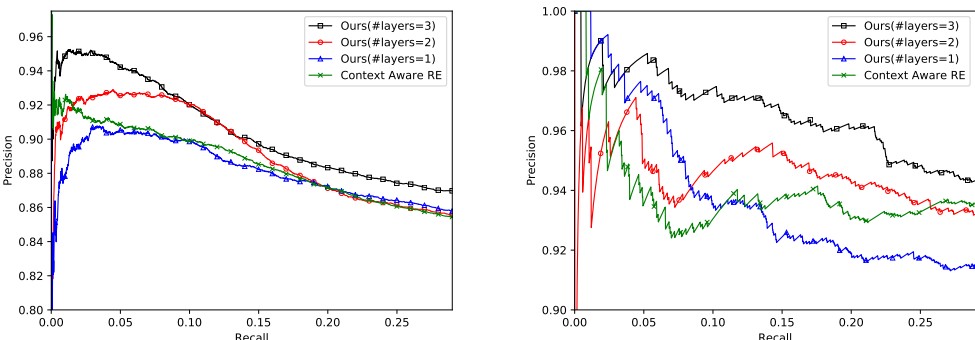

Figure 3: The aggregated precision-recall curves of our models with different number of layers on distantly labeled test set (left) and dense distantly labeled test set (right). We also add Context Aware RE for comparison.

| Sentence | Context Aware Relation Extraction | LSTM | GP-GNN (#layers = 3) | Ground Truth |
|---|---|---|---|---|
| **Oozham** ( or Uzham ) is an upcoming 2016 **Malayalam** drama film written and directed by **Jeethu Joseph** with **Prithviraj Sukumaran** in the lead role. | | | | |
| The third annual of the 2006 Premios Juventud (Youth Awards) edition will be held on July 13, 2006 at the **BankUnited Center** from the **University of Miami** in **Coral Gables**, **Florida** . | | | | |
| The association was organized in Enterprise (now known as **Redbush**) **Johnson County**, **Kentucky** in 1894 and was incorporated in 1955, after relocating to Gallipolis, **Ohio**. | | | | |

Table 4: Sample predictions from the baseline models and our GP-GNN model. Ground truth graphs are the subgraph in Wikidata knowledge graph induced by the sets of entities in the sentences. The models take sentences and entity markers as input and produce a graph containing entities (colored and bold) and relations between them. Although "No Relation" is also be seen as a type of relation, we only show other relation types in the graphs.

## 5.4 THE EFFECTIVENESS OF THE NUMBER OF LAYERS

The number of layers represents the reasoning ability of our models. A $K$-layer version has the ability to infer $K$-hop relations. To demonstrate the effects of the number of layers, we also compare our models with different numbers of layers. From Table 2 and Table 3, we could see that on all three datasets, 3-layer version achieves the best. We could also see from Fig. 3 that as the number of layers grows, the curves get higher and higher precision, indicating considering more hops in reasoning leads to better performance. However, the improvement of the third layer is much smaller on the overall distantly supervised test set than the one on the dense subset. This observation reveals that the reasoning mechanism could help us identify relations especially on sentences where there are more entities. We could also see that on the human annotated test set 3-layer version to have a

greater improvement over 2-layer version as compared with 2-layer version over 1-layer version. It is probably due to the reason that bag-level relation extraction is much easier. In real applications, different variants could be selected for different kind of sentences or we can also ensemble the prediction from different models. We leave these explorations for future work.

## 5.5 QUALITATIVE RESULTS: CASE STUDY

Tab. 4 shows qualitative results that compare our GP-GNN model and the baseline models. The results show that GP-GNN has the ability to infer the relationship between two entities with reasoning. In the first case, GP-GNN implicitly learns a logic rule $\exists y, x \xrightarrow{\sim\text{cast-member}} y \xrightarrow{\text{original language}} z \Rightarrow x \xrightarrow{\text{language spoken}} z$ to derive (Oozham, `language spoken`, Malayalam) and in the second case our model implicitly learns another logic rule $\exists y, x \xrightarrow{\text{owned-by}} y \xrightarrow{\text{located in}} z \Rightarrow x \xrightarrow{\text{located in}} z$ to find the fact (BankUnited Center, `located in`, English). Note that (BankUnited Center, `located in`, English) is even not in Wikidata, but our model could identify this fact through reasoning. We also find that Context-Aware RE tends to predict relations with similar topics. For example, in the third case, `share boarder with` and `located in` are both relations about territory issues. Consequently, Context-Aware RE makes a mistake by predicting (Kentucky, `share boarder with`, Ohio). As we have discussed before, this is due to its mechanism to model co-occurrence of multiple relations. However, in our model, since Ohio and Johnson County have no relationship, this wrong relation is not predicted.

## 6 CONCLUSION AND FUTURE WORK

We addressed the problem of utilizing GNNs to perform relational reasoning with natural languages. Our proposed models, GP-GNNs, solves the relational message-passing task by encoding natural language as parameters and performing propagation from layer to layer. Our model can also be considered as a more generic framework for graph generation problem with unstructured input other than text, e.g. images, videos, audios. In this work, we demonstrate its effectiveness in predicting the relationship between entities in natural language and bag-level and show that by considering more hops in reasoning the performance of relation extraction could be significantly improved.

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
