# OpenReview forum: "Graph Neural Networks with Generated Parameters for Relation Extraction"
_ICLR.cc/2019/Conference_

### Official Review · AnonReviewer2 · 2018-11-02
**Good results, some questions**

**Rating:** 6
**Confidence:** 4

**Review:**

This work proposes a method for parametrising relation matrices in graph neural networks (GNNs) using text. The model is applied in a relation extraction task, and specific dataset subsections are identified to test and analyse “hopping” behaviour: a model’s ability to combine multiple relations for inferring a new one.

Strengths
- strong results
- testing on both bag-level / single-level relation extraction
- Insights via multiple ablations — variation of the number of layers, exploring densely connected data subsections with cycles to identify examples for multi-hop inference
- evaluation on new, human-annotated test set


Issues
- evaluation only on one task, and one dataset (although, with more detail).
- unclear how/why the specific result metrics and their particular range of precision are chosen. Why is F1/Acc only reported for the human-labelled part, but not for the distantly labelled part? Why is precision cut off at ~25% recall? A comprehensive aggregate measure across all levels of recall would be more informative, e.g. PR-AUC., and consistently applied in all experiments.
- task varies from previous versions of the task, posing a potential problem with comparability. What is the motivation behind augmenting the dataset with “NA” labels? Why is the task from previous work altered to predicting relationships between _every_ entity pair? Also unclear: is predicting “NA” actually part of the training loss, and of the evaluation? Is training of the previous “baseline” models adapted accordingly?
- some claims appear too bold and vague — see below.
- comparatively small modelling innovation
- There is similar prior work to this, most prominently Schlichtkrull et al. 2017 [https://arxiv.org/pdf/1703.06103.pdf] who evaluate on Fb15k-237, but also De Cao et al. 2018 [https://arxiv.org/pdf/1808.09920.pdf, published within 1 month before ICLR submission deadline] who evaluate on Wikihop. These previous methods in fact do the following: “Most existing GNNs only process multi-hop relational reasoning on pre-defined graphs and cannot be directly applied in natural language relational reasoning.” It would be good to work out differences to these previous models in more detail, especially to Schlichtkrull et al. 2017.
- unclear how big the specific contribution of the language-generated matrices is. I would normally not obsess about such a baseline, but it seems that the “generate using NL” aspect is a core to the paper (even in the title), and this isn’t worked out clearly.


More comments / questions:
- “multi-hop relational reasoning is indispensable in many natural language processing tasks such as relation extraction”. This isn’t clear to me, “indispensable” is a very strong wording.
- “state-of-the-art baselines”. SOTA defines the best previous work. How can there be several baselines (plural) which are all best? What does SOTA mean when a slight redefinition of the task is proposed, as in this work?
- “Relation extraction from text is a classic natural language relational reasoning task” — reference would be useful.
- not a big issue, though this does sound somewhat contradictory: 1) “the number of layers K is chosen to be of the order of the graph diameter”. 2) “We treat K as a hyperparameter”
- not clear: is LSTM indeed exactly the same as GP-GNN with K=1? I assume there is a difference, as the LSTM encodes the entire sentence at once, conditioning entity representations on their local context, whereas in GP-GNN this would not be the case.
- The distinction to Context-Aware RE (CARE) is not clear. The authors argue that CARE models co-occurrence of multiple relations, but is this not what a multi-hop relation extraction _should_ learn? It is also not clear how GP-GNN differs in this respect.
- It would be interesting to compare with a model which does not use language to define relation matrices (A), but learns them directly as parameters (independently from the text).
- It would be interesting to see an analysis of the matrices A_{i,j}. What does the text generate, and how can one find this out?

---

> ### Author Response · Authors · 2018-11-26
> **Response to Reviewer 2 (1/2)**
>
> We thank the thoughtful advice and the following is our response.
>
> > unclear how/why the specific result metrics and their particular range of precision are chosen. Why is F1/Acc only reported for the human-labelled part, but not for the distantly labelled part?
>
> These specific metrics (PR curve, P@N%, Acc and Macro F1) are directly following previous papers (Zeng et. al 2014, 2015, Lin et. al 2015, Feng et. al 2018) in relation extraction. For relation extraction, only when the precision is high, the model is useful in practice, therefore, only high-precision part is shown.
>
> > task varies from previous versions of the task, posing a potential problem with comparability. What is the motivation behind augmenting the dataset with “NA” labels? Why is the task from previous work altered to predicting relationships between _every_ entity pair? Also unclear: is predicting “NA” actually part of the training loss, and of the evaluation? Is training of the previous “baseline” models adapted accordingly?
>
>
> In real-world applications, we do not know which pair of entities have relationships, and we need to examine every pair of entities to verify if they have relationships. It is not realistic to let the model only attend to a subset of entity pairs. Therefore, we regard our setting more realistic and correct. “NA” is a part of loss and evaluation, and baselines are trained accordingly.
>
> > There is similar prior work to this, most prominently Schlichtkrull et al. 2017 [https://arxiv.org/pdf/1703.06103.pdf] who evaluate on Fb15k-237, but also De Cao et al. 2018 [https://arxiv.org/pdf/1808.09920.pdf, published within 1 month before ICLR submission deadline] who evaluate on Wikihop. These previous methods, in fact, do the following: “Most existing GNNs only process multi-hop relational reasoning on pre-defined graphs and cannot be directly applied in natural language relational reasoning.” It would be good to work out differences to these previous models in more detail, especially to Schlichtkrull et al. 2017.
>
>
> We have added more discussions about Schlichtkrull et al. 2017 and De Cao et al. 2018 to our latest PDF version. Note that although these approaches apply GNNs to natural languages processing tasks, they still apply GNN to predefined graphs: knowledge graphs (Schlichtkrull et al. 2017), co-occurrence and co-reference graphs (De Cao et al. 2018). In sharp contrast, our model could apply directly to natural language sentences instead of predefined graphs.
>
> > “state-of-the-art baselines”. SOTA defines the best previous work. How can there be several baselines (plural) which are all best? What does SOTA mean when a slight redefinition of the task is proposed, as in this work?
>
>
> SOTA here means the best models on other relation extraction tasks. We admit that this might cause confusion, and we have removed SOTA statement in abstract and introduction in our latest PDF version.

---

> ### Author Response · Authors · 2018-11-26
> **Response to Review 2 (2/2)**
>
> > not a big issue, though this does sound somewhat contradictory: 1) “the number of layers K is chosen to be of the order of the graph diameter”. 2) “We treat K as a hyperparameter”
>
> In general graph, if K >= order of the graph diameter, then all nodes could receive global message. In our fully-connected graph, K >= 1 is enough. We treat K as a hyperparameter and show that as K grows, the reasoning ability is improved.
>
> > not clear: is LSTM indeed exactly the same as GP-GNN with K=1? I assume there is a difference, as the LSTM encodes the entire sentence at once, conditioning entity representations on their local context, whereas in GP-GNN this would not be the case.
>
>
> As we let the head entity and the tail entities carry special initial embeddings, the embeddings of head and tail entities are conditioned only on the LSTM output encoding their relationship. Hence, GP-GNN (K=1) is the same model as LSTM.
>
>
> > The distinction to Context-Aware RE (CARE) is not clear. The authors argue that CARE models co-occurrence of multiple relations, but is this not what a multi-hop relation extraction _should_ learn? It is also not clear how GP-GNN differs in this respect.
>
> Context-Aware RE is prone to errors in utilizing contextual information. Consider two relation chains in a sentence: A--R1-->B--R2-->C, D--R3-->C. When applying Context-Aware RE to reason the relationship between A and C, it attends to R1, R2 and R3, despite it has nothing to do with the reasoning chain between A and C. If exists R4, (R1, R3) ⇒ R4, the relationship between A and C is easily mistaken as R4. We have also shown in the case study that CARE is also influenced by unrelated relations. This problem limits the ability of CARE to do reasoning. It is true that multi-hop reasoning learns also the co-occurrence of relations. However, our model models the co-occurrence along reasoning chains.
>
> > It would be interesting to compare with a model which does not use language to define relation matrices (A), but learns them directly as parameters (independently from the text).
>
> The reason why we proposed to generate parameters (transition matrices) of GNNs is because “traditional” GNNs cannot be directly applied in the reasoning tasks of natural languages. Existing attempts to encode relational information in transition matrices (e.g. Schlichtkrull et al.) only learn matrices for a fixed set of relations in a predefined graph. However, for relational reasoning in natural languages (or images), the graph to be reasoned is not predefined and the relations are extracted from natural language sentences, these methods still cannot be applied.

---

### Official Review · AnonReviewer3 · 2018-11-03
**The paper describes a Graph NN method for information extraction from sentences. Some interesting innovations in the paper. The evaluation analysis and the comparison with other models are still preliminary.**

**Rating:** 6
**Confidence:** 3

**Review:**

The paper describes a Graph NN method for information extraction from sentences. The objective is to label couples of entities (token or multiple tokens)  according to a given set of relations. The GNN processes token representations through one or more layers and a final classification layer scores the relations between entity couples in the sentence. Parameters of GNN – transition matrices between layers operating on node representations – are learned from the data.  Experiments are performed on different variants of an extraction corpus, for the task consisting in extracting all the relations between identified token couples in a sentence.

The description is clear. The main original contribution is the scheme used for learning the transition matrices between layers from the text input.  Overall, the proposed system is a new mechanism for classifying relations between items in a sentence using GNN. Note that the current title is misleading w.r.t. this goal.
The authors claim that the model allows relational reasoning on text. This is somewhat exaggerated, the model performs relation classification and that’s it. There is nothing indicating any form of reasoning and this argument could be removed without reducing the value of the paper.

The experiments show that the proposed model outperforms, on this classification task baselines, including a recent model. The analysis here should be developed and improved: as it is, it does not provide much information.
Below are some questions concerning this evaluation.
Why not using mono sentence evaluation on the two distantly labeled datasets?
The performance of the two CNN baselines on the hand labeled dataset are extremely low, what is the reason for that?
The improved performance of the proposed model w.r.t. the baselines is attributed to the “reasoning” potential of the model, but this explanation falls short. There is no fact in the paper to support this idea, and the Context-aware RE model making use of sentence attention has also the potential of exploiting contextual information and thus might also infer more complex relations. The reason for the improvement has to be found somewhere else.
Overall, there is some interesting innovation in the paper. The evaluation analysis and the comparison with other models are still preliminary.

---

> ### Author Response · Authors · 2018-11-26
> **Response to Reviewer 3**
>
> We thank the thoughtful advice and the following is our response.
>
>
> > Why not using mono sentence evaluation on the two distantly labeled datasets?
>
> Mono sentence evaluation is usually not performed on the distantly supervised datasets as many instances in the distantly supervised dataset are not correct. The conventional practice is to perform bag-level evaluation on it.
>
> > The performance of the two CNN baselines on the hand labeled dataset are extremely low, what is the reason for that?
>
> As Zhang et al. 2015 pointed out, “for sentences with long-distance relations, the RNN model exhibits clear advantages”. Sentences in Wikipedia are much longer than the ones from New York Times corpus (where CNN-based approaches mostly experiment on). For bag-level RE, as long as one instance in the bag is recognized correctly and get the high probability, the bag is recognized correctly. Instead of tackling long-term dependencies, CNNs could achieve a decent result by recognizing short sentencing correctly. However, for instance-level RE, if CNNs make mistakes on longer sentences, there are no shorter sentences it can rely on.
>
> > The improved performance of the proposed model w.r.t. the baselines is attributed to the “reasoning” potential of the model, but this explanation falls short. There is no fact in the paper to support this idea, and the Context-aware RE model making use of sentence attention has also the potential of exploiting contextual information and thus might also infer more complex relations. The reason for the improvement has to be found somewhere else.
>
> Context-Aware RE is prone to errors in utilizing contextual information. Consider two relation chains in a sentence: A--R1-->B--R2-->C, D--R3-->C. When applying Context-Aware RE to reason the relationship between A and C, it attends to R1, R2 and R3, despite it has nothing to do with the reasoning chain between A and C. If exists R4, (R1, R3) ⇒ R4, the relationship between A and C is easily mistaken as R4. We have also shown in the case study that CARE is also influenced by unrelated relations. This problem limits the ability of CARE to do reasoning. We also show that by adding more layers, the performance is significantly improved, which means the multi-hop reasoning mechanism does help in our model.

---

### Official Review · AnonReviewer1 · 2018-11-03

**Rating:** 4
**Confidence:** 4

**Review:**


This paper proposes a new model based on graph neural networks for relation extraction and evaluates it on multiple benchmarks and demonstrates its ability to do some level of “multi-hop relational reasoning”.

Although the empirical results look good and the paper might present some interesting and effective ideas, I find the paper very difficult to follow and many concepts are confusing (and even misleading).

Section 3-4:

- “l” is not defined -- I assume it denotes the number of tokens in the sentence but |s| is used in other places.
- Are “entires” and “entities” the same?
- a series of tokens => a sequence of tokens.
- Equation (5): “n” is not defined on the right of equation. Does this mean that different layers have LSTMs/MLPs with separate sets of parameters? If it is the case, why do you need the word embeddings at all the layers to construct the transition matrices? Please clarify.
- Does BiLSTM return one vector or a sequence of l vectors? Even MLP needs to be defined.

In general I find the concept of “generated parameters” even confusing. How would traditional GNNs work in this context? Isn’t the novelty that parameterizing word/positional information in the transition matrix which enables a graph-based neural network working?

It would be very important to explain the intuition of this model and make the presentation clear and understandable. I don’t recommend this paper to be accepted in the current format.

The empirical results also make me wonder whether this model outperforms other models because the other models work on a single pair of entities while this model attempts to work on all pairs of entities at the same time so that it enables some level of reasoning at the entity level (e.g., language + cast member -> language spoken in Figure 1). If this is the real contribution, the paper has to make it clear enough.

Another related paper that needs to be cited:

- Zhang et al, 2018: Graph Convolution over Pruned Dependency Trees Improves Relation Extraction. EMNLP.

---

> ### Author Response · Authors · 2018-11-26
> **Response to Reviewer 1**
>
> We thank the thoughtful advice and the following is our response.
>
>
> > - “l” is not defined -- I assume it denotes the number of tokens in the sentence but |s| is used in other places. Are “entires” and “entities” the same? a series of tokens => a sequence of tokens.
>
> We have changed the notations and words accordingly. The original thought of making these distinction is to highlight the difference between GP-GNNs model and relation extraction application. The new version is more consistent.
>
> > Equation (5): “n” is not defined on the right of equation. Does this mean that different layers have LSTMs/MLPs with separate sets of parameters? If it is the case, why do you need the word embeddings at all the layers to construct the transition matrices? Please clarify.
>
> Thanks to indicating the minor mistake in our equation, and we have added “n” to the RHS of Eq. 5. In our experiments, we have tried using different and the same set of parameters in LSTMs and MLPs among layers and the results show that using different sets is better. Word embeddings in these layers are shared.
>
> > Does BiLSTM return one vector or a sequence of l vectors? Even MLP needs to be defined.
>
> We concatenated the tail hidden state of the forward LSTM and the head hidden state of the backward LSTM together to form a single vector instead of a sequence of vectors. We clarified it in the revised version to make it easier to understand. MLP is also formally defined.
>
> > In general I find the concept of “generated parameters” even confusing. How would traditional GNNs work in this context? Isn’t the novelty that parameterizing word/positional information in the transition matrix which enables a graph-based neural network working?
> It would be very important to explain the intuition of this model and make the presentation clear and understandable. I don’t recommend this paper to be accepted in the current format.
>
>
> The reason we proposed to generate parameters (transition matrices) of GNNs is because “traditional” GNNs cannot be directly applied in the reasoning tasks of natural languages. Existing attempts to encode relational information in transition matrices (e.g. Schlichtkrull et al.) only learn matrices for a fixed set of relations in a predefined graph. However, for relational reasoning in natural languages (or images), the graph to be reasoned is not predefined and the relations are extracted from natural language sentences, these methods still cannot be applied. Therefore, yes, “parameterizing word/positional information in the transition matrices” is the novelty.
>
> > The empirical results also make me wonder whether this model outperforms other models because the other models work on a single pair of entities while this model attempts to work on all pairs of entities at the same time so that it enables some level of reasoning at the entity level (e.g., language + cast member -> language spoken in Figure 1). If this is the real contribution, the paper has to make it clear enough.
>
>
> Our model is not the only one working on all of the pairs at the same time. In fact, Context-Aware RE, one of the baseline methods, also work on all the pairs simultaneously, but our model still outperforms them including Context-Aware RE.  We thus attribute its success to the superior relational reasoning ability of Graph Neural Networks.

---

### Meta-Review · Area_Chair1 · 2018-12-16
**an interesting direction but not ready for publication yet**

**Confidence:** 4
**Recommendation:** Reject

**Metareview:**

+ experiments on an interesting task: inferring relations which are not necessarily explicitly mentioned in a sentence but need to be induced relying on other relations
+ the idea to frame the relation prediction task as an inference task on a graph is interesting

- the paper is not very well written, and it is hard to understand what exactly the contribution is. E.g., the authors contrast with previous work saying that previous work was relying on pre-defined graphs rather than inducing them. However, here they actually rely on predefined full graphs as well (i.e. full graphs connecting all entities).   (See questions from R1)

- the idea of predicting edge embeddings from the sentence is an interesting one. However, I do not see results studying alternative architectures (e.g., fixed transition matrices + gates / attention), or careful ablation studies. It is hard to say if this modification is indeed necessary / beneficial.  (See also R3, agreeing that experiments look preliminary)

- Extra baselines? E.g., what about layers of multi-head self-attention across entities? (as in Transformer). What about the number of parameters for the proposed model? Is there chance that it works better simply because it is a larger model? (See also R3)

- evaluation only one dataset (not clear if any other datasets of this kind exist though)

Overall, though I find the direction and certain aspects of the model quite interesting, the paper is not ready for publication.